# Big Data Research in Fighting COVID-19: Contributions and Techniques

**Dianadewi Riswantini** *,† , **Ekasari Nugraheni** † , **Andria Arisal** , **Purnomo Husnul Khotimah** , **Devi Munandar** **and Wiwin Suwarningsih**

Research Center for Informatics, Indonesian Institute of Sciences, Bandung 40135, Indonesia; ekasari.nugraheni@lipi.go.id (E.N.); andria.arisal@lipi.go.id (A.A.); purn001@lipi.go.id (P.H.K.); devi.munandar@lipi.go.id (D.M.); wiwin.suwarningsih@lipi.go.id (W.S.)
* Correspondence: dianadewi.riswantini@lipi.go.id
† These authors contributed equally to this work.

**Abstract:** The COVID-19 pandemic has induced many problems in various sectors of human life. After more than one year of the pandemic, many studies have been conducted to discover various technological innovations and applications to combat the virus that has claimed many lives. The use of Big Data technology to mitigate the threats of the pandemic has been accelerated. Therefore, this survey aims to explore Big Data technology research in fighting the pandemic. Furthermore, the relevance of Big Data technology was analyzed while technological contributions to five main areas were highlighted. These include healthcare, social life, government policy, business and management, and the environment. The analytical techniques of machine learning, deep learning, statistics, and mathematics were discussed to solve issues regarding the pandemic. The data sources used in previous studies were also presented and they consist of government officials, institutional service, IoT generated, online media, and open data. Therefore, this study presents the role of Big Data technologies in enhancing the research relative to COVID-19 and provides insights into the current state of knowledge within the domain and references for further development or starting new studies are provided.

**Keywords:** Big Data; COVID-19; research contribution area; analytical technique; data source; survey

## 1. Introduction

Currently, the world faces many daunting challenges. In the global history of the last century, the COVID-19 pandemic and World War II have been the most severe humanitarian catastrophes. The COVID-19 outbreak is an acute respiratory syndrome and was declared a pandemic on 11 March 2020 by the World Health Organization (WHO) [1]. Furthermore, the outbreak first occurred in Wuhan in December 2019 and continues to spread rapidly throughout mainland China and worldwide, causing panic and significant losses to people's lives and economies. The virus is transmitted through direct person-to-person contact and has caused many deaths.

The COVID-19 pandemic has impacted the world for more than a year. Many countries have issued various policies to control the spread, such as working from home, learning from home, lockdown, travel restrictions, limiting the number of people in public places, and other policies [2–9]. Furthermore, it created a new standard in society, such as frequently wearing masks, washing hands, and maintaining a physical distance. This condition certainly affects almost all aspects of life, especially healthcare, social, environmental, economic, and business areas. Digital transformation programs were accelerated by different organizations and businesses during the pandemic [10–12]. In addition, online shopping and cashless transactions to avoid physical contact have now become a necessity. The daily activities of meetings, lectures, graduations, seminars, or conferences are also

held online to prevent the spread. The pandemic has also affected the environment by reducing air pollution [8]. Furthermore, the lockdown and work from home policies make many people prefer to stay at home, reduces traffic on the roads, and improves air quality in urban areas. Moreover, people prefer to ride bicycles over public transportation to avoid close contact among passengers on the local trip [13].

The war against COVID-19 is conducted by paramedics and volunteers at the forefront. Furthermore, different studies are conducted to combat and find a solution to this deadly pandemic. Many opportunities have been provided to offer technology-based solutions [14–17]. More than one year of study on Big Data for COVID-19 showed that this technology has contributed to case tracking, epidemic surveillance, virus spread and human mobility monitoring, precautionary measures, medical treatment, and drug developments [18–20]. Furthermore, advanced technology and architectures have encouraged Big Data to solve various life problems and are unavoidably utilized to cope with the pandemic. The analysis on social media related to COVID-19 contributes to solving social life problems in gaining public opinion, concern, and response to the policies implemented [18–23].

The next part of the study contains several sections, starting with a methodology of literature review process and analysis. The review on Big Data related to COVID-19 will be described in three aspects. The first is the contribution areas targeted by the study. Furthermore, previous studies were clustered into five areas based on the reviewed articles. These include healthcare, social life, government policy, business and management, and the environment described in the "Research Contribution Area" section. The literature review was based on the technology offered by Big Data coping with the pandemic described in the "Analytical Techniques" section. Generally, several methods and techniques used in Big Data technology related to COVID-19 were reported. The section explained the methods and techniques along with the application built and the analysis conducted. Finally, the types of data sources and datasets used to support the application and analysis of Big Data were described in the "Data Source" section. The conclusions of all reviews were provided at the end of this article.

## 2. Methodology

During the COVID-19 pandemic, there has been an enormous growth of data [24] that present various challenges to keep up with research knowledge within the domain of Big Data technologies [25]. Hence, this study tries to fill this gap by exploring the Big Data research for COVID-19 to identify the current research status. Similar studies were discussed by Shorten et al. [26], focusing on existing deep learning methods and how the models can provide solutions. Meanwhile, Bragazzi et al. [27] discussed possible applications of artificial intelligence and Big Data. Compared to previous studies, our study provides a broader perspective of Big Data technologies covering the application in several areas, the analytical methods, and data management. In addition, there has been no exhaustive survey within this domain.

### 2.1. Literature Review Process

Relevant and academic studies identified the application of Big Data technology in coping with the COVID-19 pandemic and to understand the current state of the study. In the analysis, articles from the Scopus citation database were selected with search key terms "Big Data" and "COVID-19" published from December 2019 until January 2021. Furthermore, the selection was conducted with the inclusion criteria of journals and proceeding articles in the English language. Qualitative analysis based on the abstract was applied to the second criteria excluding articles that were not firmly related to the context. Full-text reading was conducted to classify the results into research and review articles. Thereafter, non-empirical and less firm articles related the domain were excluded. Finally, 98 academic studies were filtered. Articles discussing empirical studies that applied Big Data analytics were grouped into research articles. Out of the total articles, 92 were

classified under research, while the rest were identified as review. Those 92 research articles are the subject of this survey study. The systematic literature review process is presented in Figure 1. The majority of research articles in the study was published in peer-reviewed journals and peer-reviewed conference proceedings. Figure 2 presented the distribution of selected research articles across journals and proceedings publications. Journals that published one article are grouped and labeled as "Various Journals" and proceedings were labeled as "Various Proceedings".

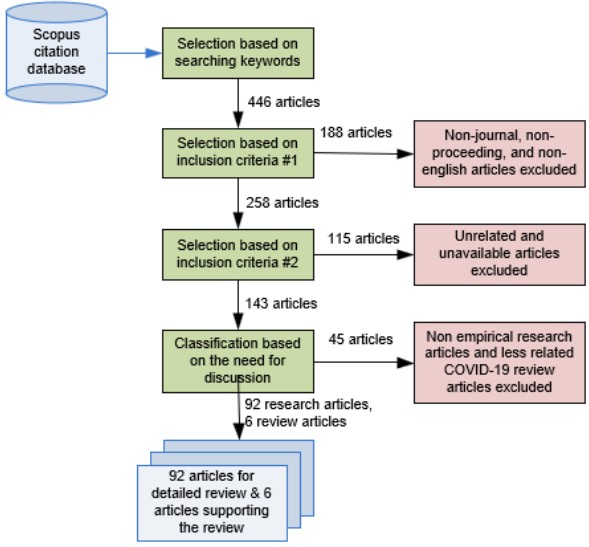

**Figure 1.** Literature review process.

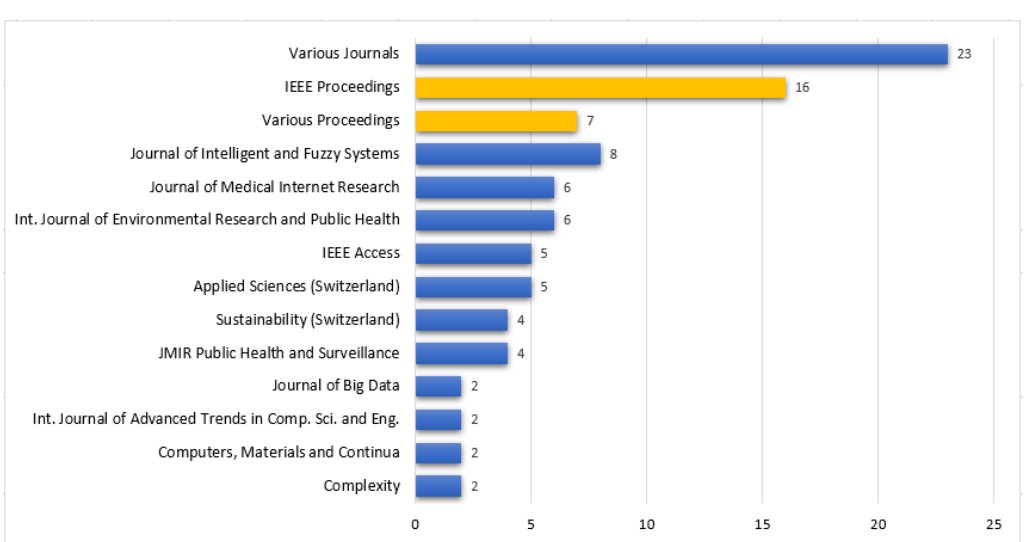

**Figure 2.** Distribution of the selected research articles across journals and proceedings.

### 2.2. Literature Analysis

After selecting the articles, a preliminary analysis for the overview of the topic concerned was conducted. Word Cloud was applied during abstracts collection to determine the occurrence and dominance of words [28]. This approach was adopted to obtain the dominant topics of the articles reviewed. The world cloud techniques on the dataset containing all abstracts of all reviewed articles were also applied. Figure 3 showed that the words health, pandemic, disease, technology, model, and analysis were dominant. For the preliminary examination, it was reported that previous studies explored more on the topic of healthcare technology. China was the most-mentioned country in the articles.

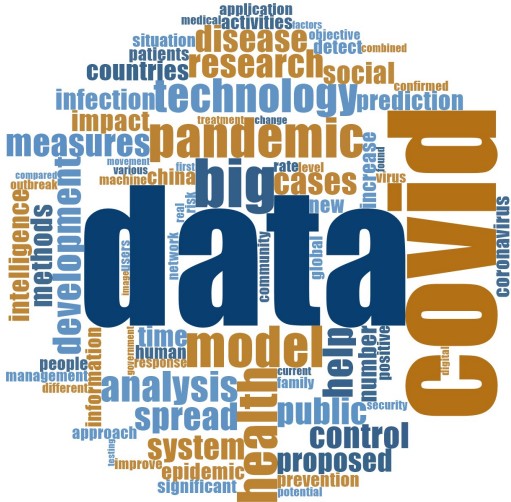

**Figure 3.** Word cloud of abstracts collection.

Furthermore, the relationship and the co-occurrence among keywords were analyzed and semantic network analysis was applied. The result is presented in Figure 4 and it showed that artificial intelligence, machine learning, and deep learning were the most used Big Data analytics methods mentioned in the keywords. Meanwhile, surveillance, infoveillance, and infodemics appeared with high dominance and the terms that describe continuous activities comprised systematic data collection, data analysis, and data interpretation towards an event related to health. These activities are related to public health measures in reducing morbidity and mortality.

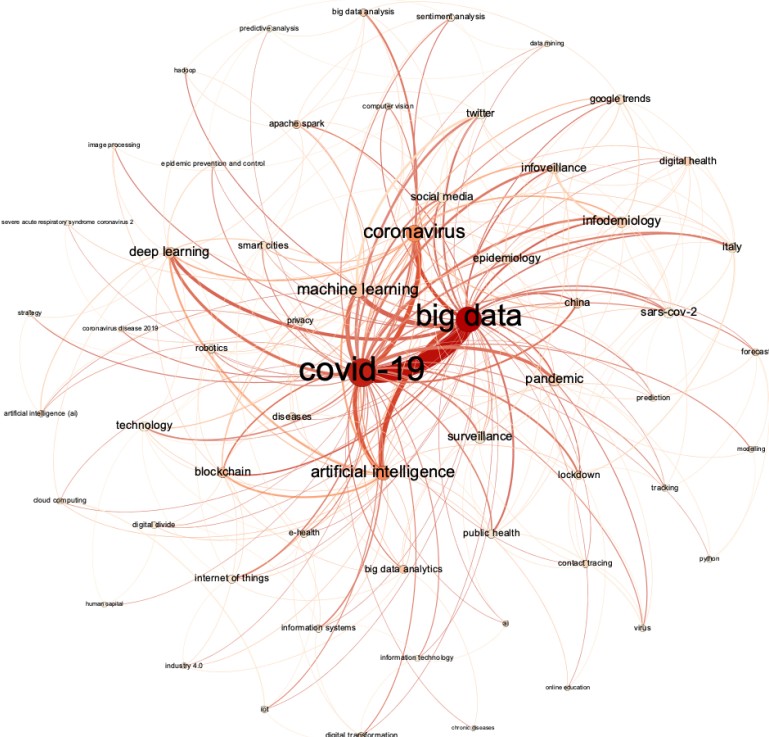

**Figure 4.** Keywords co-occurrences.

## 3. Research Contribution Area

The articles were reviewed based on their objectives and classified into health care, social life, business and management, government policy, and the environment. The classification consists of heuristics employing rational judgment based on the content of the articles. The process reduced the complexity without compromising accuracy [29]. Table 1 showed the descriptions of contribution areas.

**Table 1.** Contribution areas.

| Contribution | Covered Areas |
|---|---|
| Health care | Medical science, medicine and pharmacology, epidemiology, and health care services. |
| Social life | Behavioral sciences, psychology, and social change |
| Business and management | Hospitality and tourism, transportation, and finance. |
| Government policy | Public policy and strategy |
| Environment | Environment pollution and climate change. |

### 3.1. Healthcare

Research and development have leveraged advances in data science and Big Data technology to predict future events. Various studies related to virus transmission were carried out to predict (a) the spread of the virus [30–34]; (b) the person suspected of being infected [35]; (c) new infection areas [36]; (d) the likelihood of the second and third waves of the epidemic [37]; (e) COVID-19 contamination scenario based on people movement [38]; and (f) the increased number of cases [39].

Controlling the pandemic is key to preventing the disease from spreading further. Official data sources issued by the Government or agencies were used to capture the evolutionary trajectory of COVID-19 [40], analyze infodemiology data for surveillance [41], formulate case patterns [42], and arrange appropriate quarantines activities [43]. Furthermore, health insurance data can also be used to analyze the risk of being exposed [44]. Monitoring in public facilities prone to the transmission of the disease was also considered. This is because disease transmission in multi-modal transportation networks can be estimated using traffic flow data and COVID-19 cases [6]. Therefore, the density of transport passengers should be monitored and controlled for this purpose [45].

Previous studies attempted to improve the speed and accuracy of medical diagnostics and to find the best treatment methods for patients [46]. A diagnostic tool was developed for the early detection based on radiological images (pneumonic and non-pneumonic X-rays) [47,48]. In addition, Izquierdo et al. [49] employed a combination of some clinical variables to predict whether COVID-19 patients require ICU admission.

Studies aimed to find effective treatments without side effects are still ongoing in pharmacology and medicine. Analysis of chloroquine derivatives showed improving clinical outcomes and the reduction of mortality in COVID-19 patients [50]. Additionally, data from the Korea National Health Insurance Service showed that patients taking medication for high blood pressure have a lower risk of exposure [51].

Smart medical technology can be applied to develop IoT applications for healthcare. An application that utilizes mobile devices was designed to access information on people's health conditions dynamically. This supports healthcare professionals to monitor public health remotely. Furthermore, smart wearable gadgets can detect clinical symptoms of COVID-19 infected people [5,52]. A smartwatch can monitor their movement [53] and health parameters (such as heart rate, blood pressure, and blood oxygen), providing the signals to paramedics sent through mobile applications [38]. Previous studies showed that many infected people are asymptomatic, which can be detected using this smart technology [54].

### 3.2. Social Life

The COVID-19 pandemic has affected the economic sector and caused many social problems [55,56]. A massive amount of data available on social media were used to determine public opinion and concerns towards pandemics [57–62]. Furthermore, Big Data analytics showed the public reaction to some government policies and recommendations with respect to the lockdown policies, working from home, and social distancing guidelines [2,7,63]. User-Generated Content (UGC) in social media was extracted to detect critical events and public response to government measures in tackling the pandemic [22]. Meanwhile, social media conversations can also be utilized to expose COVID-19-related symptoms and experiences on disease recovery [64].

Moreover, the adherence to physical distancing can be monitored through a tracker device and this allows the analysis of the effect of the policies on people's activity [9]. The adherence to health protocol was inspected from the video data obtained from the camera device [65,66]. Meanwhile, the analysis of people's geolocation can provide information on human mobility changes and contact tracking [4,67–69].

Studies on COVID-19 also discussed in psychology, examining people's behavior in social situations and their capability to adapt to a particular condition [13]. Furthermore, topics in social psychology covered in the past studies include the relationship between trust and the presence of infectious disease [70]; psychological needs and their satisfaction level during the pandemic [71]; the effect of fear and collectivism on the public prevention against COVID-19 [72]; and peoples' preferences to protect the environment [73]. Some of the effects of the pandemic were studied, including family violence [21], increasing racial sentiment toward Asian people [23], the emergence of incivility and fake news on social media [30,74], and emotional tendency and symptoms of mental disorder in the face of the outbreak [75,76].

### 3.3. Government Policy

COVID-19 is a burden that drives the government to control the disease. The policies to limit community activities include working from home [2] and locking people at home to disinfect areas with high contamination levels [35]. The lockdown policy made people restrict themselves or paused their routine treatment. This is indicated by a drastic decrease in total health care expenditures based on bank transaction data [77].

The implementation of public policies needs to be analyzed to investigate the effect of the policies on the spread of disease [3]. Moreover, the government's key actions were evaluated to produce more appropriate policies for the current situation [78]. Optimization of monitoring techniques in infection areas is necessary to support the goal [79]. This is because scenario policies can differ in each region depending on the COVID-19 conditions as well as environmental and climatic factors [80]. The population-based strategies following ecological predictors were used to reduce the risk of spread [81].

### 3.4. Business and Management

The business sector has faced many obstacles during the pandemic. Chaves-Maza and Martel [82] developed a prediction model to measure the probability of entrepreneurial survival and business success based on environmental variables and public support programs. Entrepreneurs should be agile in anticipating the changes in consumer behavior. This is because the pandemic has shifted the consumer behavior and buying pattern in this uncertain business environment [74]. Furthermore, Zhang et al. [77] developed a model for figuring out health products and their utilization to obtain information regarding the customers' healthcare needs.

To survive and to stay competitive, entrepreneurs have moved to benefit off of the online channel and have enhanced their services by a product recommendation feature to improve online customer experience [83,84]. Continuous observation of product quality regarding user engagement is essential to keep businesses afloat [85–87]. Furthermore, increasing health product needs have resulted in fraud in supplying products to cus-

tomers and so the manufacturers need to fight illicit products by applying intelligent fraud detection methods [88].

The outbreak has weakened the pace of investment and the portfolio is volatile due to the effect of panic investors [89,90]. Some hold their stake while others take advantage of this situation. Sentiment analysis and time series regression can be applied to predict the future condition of the stock market [91].

The tourism and hospitality sector is impacted significantly by the pandemic. Obtaining valuable insight from data-driven analysis, tourism entrepreneurs and governments can make rational decisions to formulate the right tourism strategy and policy [92,93]. Furthermore, tourism behaviors changed in response to this new Government policy [94]. Rejuvenation of tourist areas needs to be performed by producing the existing tourism potentials that support health protocols [95]. An intelligent contact tracking system is initiated to manage tourist visits, while avoiding contact from potentially infected visitors [44,96]. In addition, passenger and traffic behaviors have also changed [97]. The changes are needed in controlling the contamination risk at the airports and on the planes as well [6,45].

*3.5. Environment*

The pandemic has affected people's way of life and their behavior towards the environment. Lin et al. [98] and Ibrahim [99] highlighted the meteorological factors that influenced coronavirus transmission. The environmental predictors were determined by surveillance of the infected areas [81]. Spatiotemporal data can reveal the distribution pattern of PM2.5 air pollution during the pandemic [100]. Meanwhile, the exposure of PM2.5 and its advancements can be used to assess the potential health risk [101]. Yan [102] proposed a reference model to prevent and control river pollution by applying microbial treatment technology using Big Data analytics.

During the pandemic, lockdown policy and mobility restrictions reduced road traffic globally. A study on Big Data quantified the impact of this traffic reduction on air quality based on meteorological and road mobility observations [8]. The data of road traffic reduction were used to predict energy consumption [103]. The outbreak has changed people's behavior towards choosing healthier transportation. Shang et al. [13] stated that the use of bikes increases the environmental benefits regarding emission reduction and energy conservation .

**4. Analytical Techniques**

The study explored the advancing Big Data technology in fighting the COVID-19 pandemic. This section highlighted the computational methods that can assist in highlighting the current and possible future state of the virus and predict the socio-economic impact on people as well as the society. It was revealed that machine learning, deep learning, and statistical algorithms were the most used methods in the COVID-19 studies. Data mining approaches used in previous studies include regression, classification, clustering, association, and social network analytics. Meanwhile, descriptive and inferential statistical analyses were also used. The special issue of the SIR (Susceptible, Infected, and Recovered) model of disease spread was discussed followed by IoT and other Big Data applications. Figure 5 presented the methods previously used concerning the underpinning applications.

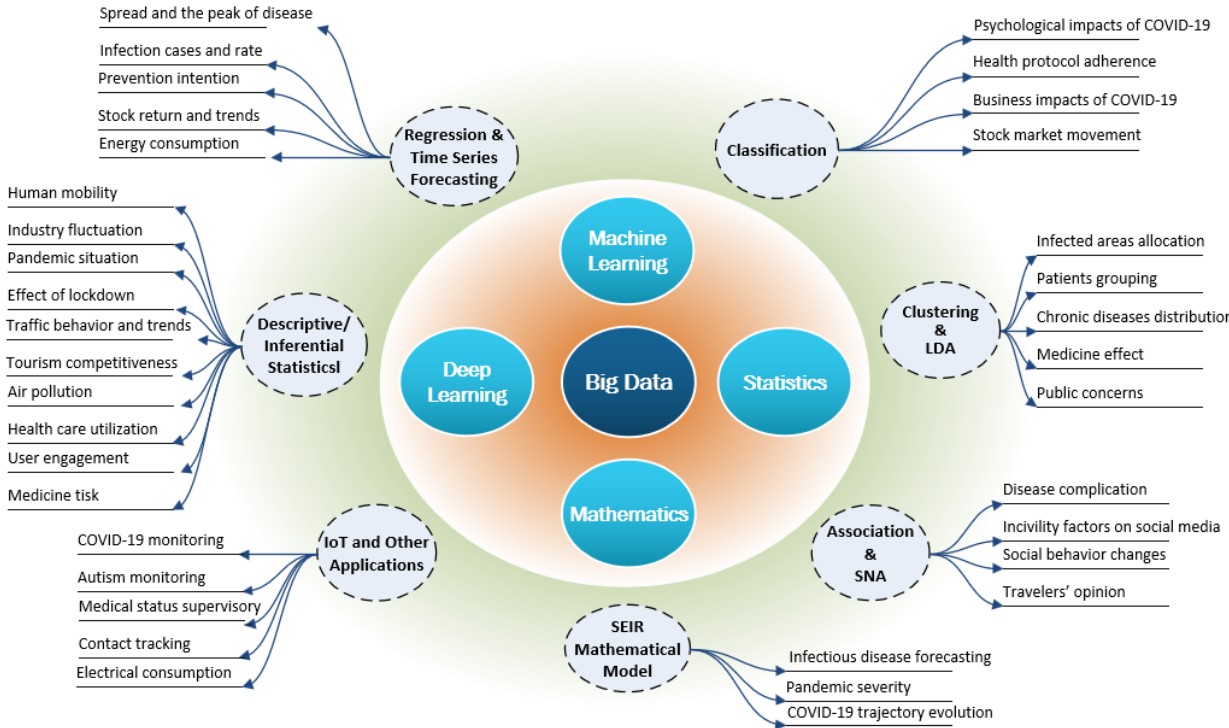

**Figure 5.** Knowledge mapping of analytical techniques.

## 4.1. Classification

Classification is a supervised learning approach which produces a model for determining an individual that belongs to a particular class. Regarding the COVID-19 studies, deep learning using techniques of RNN (Recurrent Neural Network) and LSTM (Long Short Term Memory) were used to classify the Pulmonary Function Test (PFT) image data for disease detection [48]. To determine suspected cases and areas, cell-phone spatio-temporal data were processed using a decision tree algorithm [33,36]. Furthermore, a Big Data application was developed to determine the diagnosis and treatment of the COVID-19 disease for high-risk groups combining several algorithms such as Extreme Learning Machine (ELM), Generative Adversarial Networks (GANs), deep learning techniques RNN and LSTM using clinical data and medical images [46].

Sun et al. [76] developed a psychological computing model to identify the continuous emotional symptoms of mental disorders. This mental health recognition application performs visual analysis and considers speech and facial expression images as multi-modal data. In addition, it explores a relationship between short-term basic emotions and long-term complex emotions. This emotion-sensing model used bi-directional LSTM and three-Dimensional CNN. Furthermore, people's psychological needs were observed by analyzing user-generated content posted on Twitter. Long et al. [71] applied Natural Language Processing (NLP) and Support Vector Machine (SVM) to study this subject. Moreover, a similar technique was utilized to investigate the shifts in anti-Asian racial sentiment regarding the emergence of COVID-19 [23]. Mackey et al. [64] conducted an infoveillance study on Twitter and Instagram to expose counterfeit health products and characterized the information in terms of product types, selling claims, and sellers types by combining Fine-tuned pretrained LSTM and Bi-Term Topic Modeling.

A computer vision application, which detects objects and distances, was developed using the Kubeflow machine learning platform and OpenCV library. This study analyzed crowd conditions from the video streaming data [65]. In attempting to monitor and enforce the health protocol adherence, an application of face recognition was developed

by adopting CNN (Convolution Neural Network) to determine when someone is wearing a mask or not [66].

A classification learning technique of MLP (Multi-Layer Perceptron) may be applied to predict the resilience of entrepreneurs facing the pandemic. Five clusters were categorized into the three classes of success, survive, and fail using SOM (Self-Organizing Map) [82]. Furthermore, CNN was applied to determine the industry category based on the economic indicators using a single and hybrid database [92]. Sentiment analysis complemented with regression was used to predict the stock market movements during the pandemic [91,104].

### 4.2. Clustering and Topic Modeling

As an unsupervised learning approach, the clustering groups entities based on their similarity. K-means algorithms integrated with correlation techniques may be employed to cluster the countries based on the pandemic stages and to examine the relationship between public policies and the spread of disease [3]. Shahata et al. [36] used K-means clustering to allocate positive case areas and to classify the risk status using decision trees algorithms. The K-modes clustering algorithm was used to group the patients to analyze their health and the necessary treatments. Then, chronic disease distribution among clusters can be explored [105]. K-means clustering was employed to allocate infected areas to classify a person's risk [45] and identify the spreading of coronavirus [33]. Hierarchical clustering was applied to identify the actual groups of infected patients [79] and the effects of chloroquine derivatives [50].

The Bi-Term Topic Model (BTM) was applied to analyze Twitter micro-blogging (tweets) while identifying the pros and cons of the government's social distancing guidelines. Combined with social network analysis, the study investigated the networked structure of the Twitter communication dynamics [63]. Furthermore, a survey on public opinions on the remote working policy used the K-means algorithm to cluster the posted tweets [2]. Some studies showed hidden themes from the tweets to explore the public concern about pandemic issues using Latent Dirichlet Analysis [22,106].

### 4.3. Association and Semantic Network Analysis

Association is a form of unsupervised learning that aims to find the relationship between entities from a large dataset. The application for COVID-19 was conducted using the Frequent-Pattern growth (FP-growth) algorithm to analyze the relationship among various diseases and the associated complication problems [54]. Almasmani et al. [84] developed an association rule algorithm based on the cosine similarity to identify customers' shopping behavior by examining associations between items purchased on their shopping cart.

Generally, Semantic Network Analysis (SNA) is used in text mining to analyze social media data. A study on figuring out the incivility factors on social media was conducted using mixed SNA with binary logistic regression classification [58]. Sung et al. [94] employed SNA to explore travelers' perceptions and interests after the extensive spread of COVID-19. Centrality and convergent correlation were equipped for this semantic network analysis.

### 4.4. Regression and Time Series Forecasting

Regression is used to estimate value and to determine the causal relationship of a set of variables. In comparison, time series forecasting is a technique for the prediction concerning the time sequence, analyzing past trends, and assuming that future and historical trends will be similar. A study on COVID-19 applied a regression model to predict infected cases and was compared with ANN prediction used to indicate the spread and the peak number of COVID-19 cases [32]. Furthermore, differential private ANN was developed to make predictions with the feature of individual data privacy protection. This extended model proved that introducing Laplacian noise at the activation function level produced results similar to the base ANN [107]. A study on the spread prediction was performed by creating an ensemble model from the decision tree and logistic regression used to

develop a tree-based regressor model for higher accuracy [31]. Ye and Lyu [70] studied the impact of trust and risk perception on the infection rate using multilevel regression for the city and province-level analysis. Furthermore, multiple regression was adopted to observe the preventive intention based on social media data. The result showed that fear and collectivism positively impacted the community prevention intentions but reduced positive influence among persons [72].

Lee [91] exploited the impact of COVID-19 sentiment on the US stock market differentiated by industries. The study developed time series regression models and used the data from Google Trends on coronavirus key-term and daily news sentiment index for the analysis. Meanwhile, a study on the stock market employed a regression model to reveal the impact of investor attention and the number of media reports about masks on the 40 mask concept stocks' rate of return [90]. Several studies on time series prediction were conducted for energy and electricity consumption forecasting [103,108].

### 4.5. Descriptive and Inferential Statistics

The study of human mobility during the pandemic was conducted by considering three fundamental metrics; number of trips per person, person-miles traveled, and proportion of staying home. Based on these metrics, the effect of policies across regions under diversified socio-demographics was observed. Also, a Generalized Additive Mixed Model (GAMM) was generated for inferential analysis [67]. Concerning human mobility, flight traffic behavior was monitored for countries to examine the relationship between the number of flights and the COVID-19 infection employing descriptive statistics [97]. The descriptive analysis was expanded with repeated measures through analysis of variance (ANOVA). Meanwhile, the correlation analysis was implemented to study the hotel industry's turbulence impacted by COVID-19 [93].

Previous studies discovered the correlation between the incidence of COVID-19 and search data provided by Google Trends, and the regression lines were derived to predict the evolution of the pandemic [37]. A similar study was conducted using Pearson correlation and ARIMA (Auto-Regressive Integrated Moving Average) to show the relation between Google Trends data and COVID-19 cases [34,41]. Descriptive statistics were further employed to exploit the effect of lockdown on people's activities represented by the number of steps per day regarding the adherence to staying at home policy [9]. Furthermore, Gualtieri et al. [8] observed the impact of road traffic on air quality in several urban areas. The analysis considered the time series of traffic mobility to show the association among meteorological parameters, road traffic, and pollutant concentrations. Some studies on the air quality, the pollution risk, and health city conditions during the outbreak were conducted using various statistical descriptive techniques [55,85,86,100].

Study on the evaluation of eco-tourism resources employed PCA's statistical technique (Principal Component Analysis) to diminish the indicators for the tourism index system. In addition, the method was integrated into the AHP (Analytical Hierarchy Process) for generating an evaluation index system of urban tourism competitiveness [95]. PCA was also applied for evaluation of online service-learning, which was distinctively raised during the outbreak. It was used to develop a user-engagement score system by applying Pearson correlation to discover the association with the number of subscribers and their reviews [87]. Another statistical analysis performed was DID (Difference-In Difference) techniques, which were employed to identify the effect of the medicine on the risk groups of COVID-19 [51] and the individual changes in health care utilization from different risk groups [77].

### 4.6. SIR/SEIR Model

The prediction and control of infectious disease spread can be analyzed using SIR model.The SIR (Susceptible, Infected, and Recovered) is a mathematical and epidemiological model which is one of the core epidemiological models for analyzing infectious disease outbreaks with more specificity in modeling population subsets for accurate fore-

casting [26]. The model can be extended to an SEIR model by including various sizes of the Exposed (E) population and more detailed data.

Wang et al. [30] compared several prediction models of the epidemic situation based on COVID-19. The models compared are SIR combined with least square, SIR combined with particle swarm optimization, and classical logistic regression. The study showed that the logistic regression model provides more in line with actual conditions than the two other models.

Liu et al. [40] developed the SEIR model for capturing the trajectory of COVID-19 evolution in Wuhan using various assumptions to evaluate how the population is exposed (E) by suspected people (S) that still stay in Wuhan. However, the model ignored the suspected people who have been moved out. Infected people (I) are distinguished into infected people who are quarantined in hospitals and not. The assumption is that the hospitalized people cannot spread the virus outside the hospital, while non-hospitalized people are most likely to spread the virus. Both types of infected people may be recovered (R) or pass away. Moreover, this model considers influencing external factors such as city closures, shelters, and additional hospitals, pandemic size, and duration to forecast the peak condition.

Eksinchol [80] developed another SEIR model to estimate pandemic conditions by adapting the actual COVID-19 data of suspected people (S) for each province in Thailand. The model extended the exposed population (E) variable into asymptomatic people and pre-symptomatic people. Both types of the exposed population would be asymptomatic or pre-symptomatic infectious people (I). The model takes the assumption that all infectious people will recover (R). The authors took the assumption because Thailand's mortality rate is relatively low (2%). Therefore, the model neglected the dead proportion in the calculation. In addition to considering the different recovery rates and transmission for each province, this model also considers the mobility factor between areas that can spread the disease to other places. In contrast to the previous model (Liu's), which paid more attention to quarantined people, whether hospitalized or not, in modeling the spread of the virus. Eksinchol's model is more concerned with the symptomatic aspect of infectious people.

### 4.7. IoT and Other Big Data Application

IoT system integrates several components, consisting of sensors/devices that send data to the cloud through several connectivity types. It provides a solution for remote monitoring and control. IoT technology for COVID-19 is mainly conducted in the health sector, and smart devices are connected to the patients to monitored their condition remotely by paramedics in real-time through a mobile application. The digital transformation for the public health care system was carried out by adopting a fog environment that integrates several local devices connecting to the cloud infrastructure. The environment can improve the quality of the data [38], and new IoT-fog-cloud-based architecture was proposed by Kallel et al. [53] to monitor for autism and COVID-19 patients. The system has several advantages, including real-time data processing, data integrity for a multi-tenant environment, and business processes running in the cloud.

Ashraf et al. [5] introduced a strategy of layered edge computing mechanisms to identify medical health status (such as fever, heartbeat, and cardiac condition) based on data collected from wearable smart gadgets. The proposed framework provided a continually updated map/pattern of the infected since the suspected can be tracked and keep safe from other people. This layered mechanism reduced the system delay factor and delivered a quick response. The system provided notifications, awareness, recommendations, and assistance on the user application layer.

Efficient control of the pandemic spread by rapidly isolating and disinfecting suspicious sites was offered by Benrequia et al. [35]. A Big Data architecture that automatically and continuously collects geolocation data from people's outdoor activities through IoT devices was also proposed. The IoT system was used to determine all individuals that have contacted infected persons through the spread trajectories.

Another application that uses a Big Data approach was smart power grids. Furthermore, a more resilient smart grid analysis through a Big Data-based approach as conducted by Bionda et al. [109] using a smart grid semantic platform. The study showed the system can manage sudden anomalies of electrical energy consumption by updating the load profile based on forecast data in the medium short term.

## 5. Data Source and Dataset

Big Data is classically characterized by "4Vs", where: (1) Velocity refers to the speed of data transfer and processing; (2) Volume points out the fact that a huge amount of data are now produced and available every time; (3) Variety represents the number of data sources in various types and formats; (4) Veracity concerns with the accuracy and the validity of data [27,110]. Big Data analytics is an advanced analytic technique used for extracting knowledge from a huge volume of data. The skyrocketing amount of data and the advance of computing technology have accelerated the applications in processing large data stored in a distributed file system.

Several applications developed concerning to COVID-19 examine massive amounts of data on several distributed servers, requiring a supporting storage system. The existence of a cloud network provide higher performance for a large dataset [111]. Furthermore, distributed NoSQL database technology has scalability, flexibility, and high performance, which is considered most suitable for processing Big Data. This database system is non-relational and can manage databases with a flexible schema and does not require complex queries. Some of the NoSQL database technologies used in the reviewed articles include Cassandra, MongoDB, Hbase, and Neo4j as shown in Table 2.

MySQL and PostgreSQL are relational database management systems, where the data search process is linear with the amount held. With a greater volume of data required, there will certainly be more time-consuming for the search process. Regarding the management of big volumes of data that the server may become overload and cause bottlenecks; the data needs to be integrated into a Big Data library framework (Apache Hadoop software library) [3,54,97]. Partitions are one of the framework's main features. The feature can distribute data to predefined partition nodes adjusted to business requirements. Hence, the query process on massive data remains reliable.

Hadoop is a Big Data framework managing distributed storage systems that enable access and processing of an immense volume of data. The principle is a cluster of nodes, where one cluster coordinates many nodes, and each node has its own data storage and processing. Furthermore, this technology provides a solution for relational databases to manage large volumes of data. The data are transferred from relational databases to Hadoop and vice versa through the Apache Sqoop (SQL(Structured Query Language) to Hadoop) tool. Apache Spark is an open-source streaming platform in the middle layer. It separates data streams and analyzes or transmits them in real-time to Hadoop Big Data lakes, applications, and systems analysis. Furthermore, Kafka is also used for high throughput and low latency stream processing from website activity tracking to real-time analytics, such as Twitter data analysis [21,59].

The literature review showed a study on Big Data related to COVID-19 and uses a wide variety of data sources available publicly or privately. The data were categorize into six classes: government official, institutional service, IoT generated, online media, public/open data, and others. Table 3 presents the dataset used in the previous research.

**Table 2.** Databases technology with their applications used by previous research.

| Technology | Description | Applications | Data Usage |
|---|---|---|---|
| Cassandra | Distributed NoSQL databases designed to handle large amounts of data spread across multiple servers | Smartwatch for monitoring system [53]. | Patient movement data from IoT Devices. |
| Databricks | Cloud-based data engineering tool used for processing and transforming massive quantities of data under Apache-Spark-based platform. | Analysis of the needs of the distribution system operator for the electricity grid [109]. | Milan's Distribution System Operator and meteorological data. |
| Hbase | An open-source non-relational distributed database system, column-oriented capable of processing large-scale-data and is built on top of the Hadoop Distributed File System (HDFS). | Video streaming data analysis [65]. | Streaming video data of people's movements in public places |
| Neo4j | An open-source graph database management system developed by Neo4j, Inc. | Insight-driven learning (IDL) for healthcare [60]. | Patient conversations tweets |
| MongoDB | Document oriented, NoSQL, cross-platform distributed database to store data in JSON-like documents | Healthcare monitoring remote system [38]. | Heartbeat, blood pressure, sleep, blood oxygen and people movement data from IoT Devices. |
| | | Classification of residents' psychological needs [71]. | Residents'tweets. |
| | | Detection of public concern [22]. | People concern, pandemic measures and daily livelihood tweets. |
| MySQL | An open-source relational database management system to store structured data under the license of GPLv2 or proprietary | Analysis of flight traffic behavior [97]. | Flight traffic data for each airport and the COVID-19 infection data for each country. |
| | | Analysis of health data [54]. | Medical data from IoT Devices. |
| PostgreSQL | An open-source relational database management system to store structured data under the license of PostGreSQL for free and open-source of permissive. | Cluster analysis to identify data patterns of the public policy implementation [3]. | COVID-19 Global Cases data from JHU, US Lockdown Dates Dataset and COVID-19 Government Measures Dataset. |

**Table 3.** Data source and dataset.

| Data Source | Dataset | References |
|---|---|---|
| Government Official Data | | |
| Public Health | COVID-19 cases and events | [32,37,40,42,43,70,77,79,80,105,112] |
| | Clinical, demographical, and laboratory data | [46,48,98,105] |
| Insurance Services | Medical prescription and Health insurance claim | [44,51] |
| Energy Consumption | Electricity and total energy consumption | [103,108,109] |
| | Daily health condition of the passenger | [45] |
| Transportation | Traffic flow and Traffic density | [6,8,13,98] |
| | Residential car park | [68] |
| Tourism | Urban tourism data | [95] |

**Table 3.** *Cont.*

| Data Source | Dataset | References |
|---|---|---|
| Institution Official Data | | |
| WHO, John Hopkins, and ECDC | COVID-19 global cases | [3,31,41,98,99] |
| Bank cards transactions | Healthcare expenditure | [77] |
| Andalucia Emprende Foundation | Entrepreneur data | [82] |
| Yale Industrial Economic | Industry statistical data | [86,92] |
| Mob-Tech Research Institute | Internet usage data | [78] |
| International hotel chain | CRM public data | [74] |
| European Society of | COVID-19 negative X-ray images | [74] |
| Radiology | Facial expression video and speech audio data | [76] |
| IoT Data | | |
| GPS data | Position location | [44] |
| Camera Video | Video streaming | [65] |
| | Human mobility and human steps records | [9,53,67,69] |
| Smart/Mobile Devices | Physical health records | [5,38,54] |
| | Geolocation of suspected/infected | [35,36,44] |
| Monitoring devices | Water quality and PM2.5 concentration | [100–102] |
| Online Media Data | | |
| Social networking service | Twitter, Weibo, Instagram, Facebook, and WeChat data | [2,4,21,30,57–60,63,71–73,78,106] |
| Navigation | Google data, Baidu data | [34,40,41,50,55,70,81,90,91,112] |
| Online news | Fox news, Korean and China-news, and magazine | [61,73,113] |
| E-commerce | Online shopping data, Tripadvisor data | [83,93] |
| Public/Open Data | | |
| Stock Exchange | Stock market data | [89,104] |
| Kaggle | COVID-19 cases | [31,33,107] |
| Scientific data | Weather and climate data | [99] |
| Other datasets | Masked face head pose image data | [66] |

WHO and JHCRC (Johns Hopkins Coronavirus Resource Center) provide data and statistics mainly used for the COVID-19 studies. The data expose the situation of viruses spread by country, territory, or area. These data monitor and conduct activities to control the spread of the virus [3,31,41]. The official data of infected, recovered, dead, regional risk zones, and distribution of cases, as well as transmission, was also officially published by the government of each country [32,37]. Transportation, tourism, and industries data have also become the concerns in the study [6,95,98].

IoT systems have generated real-time Big Data from sensing devices, such as GPS, CCTV, cameras, smart/mobile devices, and monitoring devices. The use of smart wearable devices makes personal physical health data easier to be obtained and monitored [5,9,38]. IoT studies also monitor the changes in environmental quality and energy consumption due to human behavior shifts [100–102].

Social media contributes significantly to the development of Big Data. It is the most preferred means of communication for human interaction. Micro-blogging platforms such as Twitter and Facebook are the most active social media network platform that supports the studies on public opinion [7,22,64], public concern [72,78], and psychological condition towards the pandemic [23,71,106]. Users' comments on social media can be analyzed to scrutinize people's behavioral changes due to the outbreak from many perspectives [94]. Public or open datasets are used to handle various studies on COVID-19, including the stock market, weather, and climate data.

## 6. Conclusions

The use of Big Data technology in tackling the COVID-19 outbreak was discussed. This pandemic has induced many problems in various sectors of human life. To capture the landscape of the study, reviewed articles were categorized into contribution areas in previous Big Data studies. Furthermore, methods and techniques were discussed to show the role of Big Data analytics in solving the problem and their contribution to the body of knowledge. The analytical techniques refer to computational domains, including machine learning and deep learning as well as statistical analysis. Artificial intelligence fields of computer vision, remote sensing, the internet of things, and natural language processing were tested in solving COVID-19 problems. In addition, data sources were addressed with different data types to guide future studies in developing the data-driven application.

Big Data technology has demonstrated its significant role in the COVID-19 study. Furthermore, previous studies had contributed mainly to areas of healthcare, social life, government policy, business and management, and the environment. Healthcare and social life areas received main interest. Many analytical techniques were applied for handling numerable issues, including epidemic surveillance, medical diagnostics and treatments, monitoring of health protocol, social changes, consumer behavior, and the effects of the pandemic on the earth systems.

Machine learning, deep learning, statistical, and mathematical methods, as well as their combination have been widely employed to solve pandemic issues. Machine learning and deep learning are the most frequent techniques used in Big Data analytics due to their ability to give better results with the increasing amount and variety of data. Moreover, the advances of IoT technology and smart devices generate more streaming data, which leads to the increasing implementation of Big Data analytics on data processing framework such as Hadoop and Spark and distributed non-relational databases such as HBase or MongoDB.

There are still many challenges ahead in dealing with COVID-19. The emerging new variants, vaccine effectiveness and side effects, relaxation of health protocols and new normal adaptation, medical waste management are issues to be resolved in the future. A wide range of Big Data technology provides opportunities to solve these problem challenges. Therefore, insights into current states of knowledge on Big Data technology for COVID-19 and references for further development or starting new study are provided.

**Author Contributions:** Conceptualization, D.R. and E.N.; methodology, D.R.; formal analysis, D.R. and E.N.; writing—original draft preparation, D.R. and E.N.; writing—review and editing, A.A.; visualization, D.M.; supervision, P.H.K.; project administration, W.S. All authors have read and agreed to the published version of the manuscript.

**Funding:** This research received no external funding.

**Institutional Review Board Statement:** Not applicable.

**Informed Consent Statement:** Not applicable.

**Data Availability Statement:** Data sharing not applicable. No new data were created or analyzed in this study.

**Acknowledgments:** The authors wish to thank the other members of the information retrieval research group at Research Center for Informatics, Indonesian Institute for Sciences, for their help and supportive discussions throughout this work.

**Conflicts of Interest:** The authors declare that they have no competing interests.

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
