# Peer review of "Big Data Research in Fighting COVID-19: Contributions and Techniques"

_2504-2289, doi:10.3390/bdcc5030030_

Round 1
Reviewer 1 Report
This paper is a review of current big data research with applications for Covid-19 research. The authors implement a systematic review of the literature and relate previous research on big data to current applications for Covi-19. I think the goals of this paper are laudable, and I do like the connection between past uses of big data and current applications for Covid-19.
I believe the goal of Table 2 is to relate previous research to the current Covid-19 applications. I would like to see another column in Table 2 that states what the databases are currently used for. I know you state it in the paper, but having it in the table would be helpful.
I think Figure 5 is great! I feel it is one of the best pieces of the paper as it clearly shows how the applications are being carried out.
I am not sure why you gave the SIR and SEIR models their own figures (6 - 8), should all the techniques have their own figure?
Reviewer 2 Report
Abstract)
Please, add something about the benefits of the research.
Introduction)
(…) After one year of the pandemic, many studies have been carried out to expose
1various technology innovations and applications to combat the coronavirus that has
killed numerous people. The pandemic has accelerated the use of Big Data technology to
mitigate the threats of COVID-19 (…). [Please provide references]
Research Contribution Area)
Please provide the references or/and arguments for classifing articles into health care, social life, business and management, government policy, and the environment groups.
Methodology) – The authors should identify the strong research gaps as the literature review itself and exploring the Big Data research for COVID-19 is not a scientific aim.
The authors should also provide some fields of future research regarding to identified research gaps.
Conclusions) Please syntesize the insights into current states of knowledge on Big Data technology for COVID.
Reviewer 3 Report
Please see the attached file for the comments.
